# Changing Role of Users—Innovating Responsibly in Digital Health

**Tatiana Iakovleva [1],\*, Elin Oftedal [2] and John Bessant [1]**

1 Department of Innovation, Marketing and Management, Stavanger Business School, University of Stavanger, 4036 Stavanger, Norway; johnrbessant@googlemail.com
2 Department of Media and Social Sciences, Faculty of Social Sciences, University of Stavanger, 4036 Stavanger, Norway; elin.m.oftedal@uis.no
\* Correspondence: Tatiana.a.iakovleva@uis.no

**Abstract:** Despite the recognition of the importance of stakeholder inclusion into decisions about new solutions offered to society, responsible innovation (RI) has stalled at the point of articulating a process of governance with a strongly normative loading, without clear practical guidelines toward implementation practices. The principles of RI direct us to involve the user early in the innovation process. However, it lacks direction of how to involve users and stakeholders into this process. In this article, we try to understand how to empower users to become a part of innovation process though empirical cases. Based on 11 cases of firms innovating in digital health and welfare services, we look on firm practices for user integration into their innovation process, as well as how the user's behavior is changing due to new trends such as availability of information and digitalization of services. We try to explore this question through lenses of responsible innovation in the emerging field of digital healthcare. Our findings indicate that users are not a homogenous group—rather, their willingness to engage in innovative processes are distributed across a spectrum, ranging from informed to involved and, at extreme, to innovative user. Further, we identified signs of user and stakeholder inclusion in all our cases—albeit in different degrees. The most common group of inclusion is with involved users, and firms' practices varying from sharing reciprocal information with users, to integration through focus groups, testing or collecting a more formative feedbacks from users. Although user inclusion into design space is perceived as important and beneficial for matching with market demands, it is also a time-consuming and costly process. We conclude with debating some policy impacts, pointing to the fact that inclusion is a resource-consuming process especially for small firms and that policy instruments have to be in place in order to secure true inclusion of users into the innovation process. Our article sheds light on RI practices, and we also suggest some avenues for future research to identify more precisely whom to include, when to include and at what stage of the innovation process.

**Keywords:** responsible innovation; user engagement; role of users; digital health; informed; involved; innovative patient

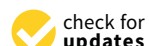

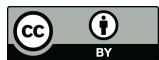

## 1. Introduction

Innovation involves creating value from ideas, but this raises the question of for whom is the value created? At a time when there is growing recognition of Global Challenges [1], responsible research and innovation (RRI) is suggested as a way to govern innovation development to address challenges populations face, such as poverty, inequality, aging population, and availability of quality healthcare. Such principles suggest a broader stakeholder inclusion into the decision-making process, anticipation of societal needs, and reflection of concerns [2], which calls for new innovation policies to enact it [3]. However, despite the recognition of the importance of stakeholder inclusion into decisions about new solutions offered to society, RRI appears to have stalled at the point of articulating a

process of governance; it often has a strongly normative loading, without clear practical guidelines toward implementation practices [4]. The principles of RRI direct us to involve the user early in the innovation process but their translation often lacks direction as to how to involve users and stakeholders in this process. The idea of "responsible innovation" has a long heritage as a field of research and practice. Today the discussion focuses on key themes such as sustainability, ethics, and social responsibility in a wide range of books and journals [5–7].

However, in our view, "responsibility" has lost meaning as a standard against which differentiation can be made between good and bad activity [4], therefore hindering progress towards generally more responsible aggregate outcomes. There is a tendency for technology actors to dominate social actors in responsibility processes, reducing the "citizen" to one common actor because of the difficulties of translating messy social voices.

Inclusion has the objective of broadening visions, purposes, issues, and dilemmas for wide and collective deliberation through the processes of dialogue, engagement, and debate [8]. Inclusion intends to develop greater democratic accountability in the innovation life cycle [9]. Von Schomberg [10] (p.1), argues that " *RRI should be understood as a strategy of stakeholders to become mutually responsive to each other and anticipate research and innovation outcomes underpinning the 'grand challenges' of our time for which they share responsibility.*" Although the calls for inclusivity are often raised in literature, there is still a limited number of empirical studies that look on how inclusiveness have been exercised in organizations trying to implement innovations [11]. There is a need to explore firms' practices to understand how organizations approach user inclusion, what challenges and opportunities that implies, and whether users can be perceived as homogenous group that should be treated equally. Moreover, in modern society, processes such as digitalization might significantly change traditional practices of user inclusion, as well as change who the users are and users' behavior [12]

In the current article, we try to understand ***how to empower users to become a part of innovation process in a responsible way.*** For that, we need a deeper understanding of how firms integrate users into their innovation practices, as well as how user behavior is changing due to new trends such as availability of information and digitalization of services. We try to explore this question through lenses of responsible innovation in the emerging field of digital healthcare and focus particularly on the challenge of increasing user engagement in the process. This article is based on empirical insights from 11 international cases of firms and their practices of user inclusion. The findings point that users should be treated as a heterogenous group in relation to their willingness to be involved into the innovation process. Further, acting responsibly and including users early on into innovation process is rewarding, but at the same time is a time-consuming and costly process. We suggest that support policies could be beneficial to ensure user and stakeholder inclusion into innovation processes. The rest of this paper is organized as following: first, we discuss inclusiveness as the driver of responsible innovation, the changing role of users due to digitalization and new opportunities it provides for user inclusion. Next, we describe the method applied to data gathering and analysis, after which we present findings from our empirical cases, organized through three user categories—informed, involved, and innovative users—and shed light on their role in innovation process. Finally, we conclude this article by outlining theoretical and policy implications, as well as discuss the future research avenues.

## 2. Inclusiveness as a Driver of Responsible Innovation

There has been extensive concept development in responsible research and innovation (RI) [4,8,13,14] but these discussions are not yet concentrated into particular fields; instead, RI is a truly cross-disciplinary debate. Despite the growing body of literature, the concept of RI is still largely normative in nature. RI's basis is that "responsibility" in an innovation emerges when societal actors have opportunities to endorse behavior that influences the innovative process in translating an idea into a launched product, service, or technique.

However, questions still remain about which stakeholders to include, at what stage, and under what circumstances [15].

Extending the RI model suggested by Stahl et al. [16], we argue that for innovation to be able to diffuse in a responsible way, its purpose, process, and outcome with regard to ethical and responsible behavior should consider four elements suggested by Stilgow, Owen and Macnaghten [6]: anticipation, inclusiveness, reflectiveness, and responsiveness. To outline, these dimensions might be described in the following way:

- Anticipation—describing and analyzing those intended and potentially unintended impacts that might arise, be these economic, social, environmental, or otherwise.
- Reflection—reflecting on underlying purposes, motivations, and potential impacts, and on associated uncertainties, risks, areas of ignorance, assumptions, questions, and dilemmas.
- Inclusivity—opening up visions, purposes, questions, and dilemmas to broad, collective deliberation through processes of dialogue, engagement, and debate, inviting and listening to wider perspectives from public and diverse stakeholders.
- Responsiveness—using this collective process of reflexivity to both set the direction and influence the subsequent trajectory and pace of innovation.

In this article, we focus on the inclusivity dimension and below we broadly discuss this construct. Inclusion refers to the involvement of different stakeholders in innovation activities to represent their ideas, creativity, and voices. The inclusion of public and all the relevant actors in the governance of science and innovation is a growing requirement for legitimacy [17,18]. This opens the platform for dialogue and discussion that provides social intelligence, which would mediate in avoiding adverse public relations [19]. Furthermore, inclusion and deliberative participation of different actors in the innovation process helps the development of perceived ownership of the innovation outcomes and motivates creativity [20]. The effectiveness of an inclusion exercise can be determined by how efficiently, complete and relevant information is obtained from all appropriate sources, transferred to, and processed by those responsible and combined to generate a response [21]. Nevertheless, over-inclusion of participants in the decision-making process, on many occasions, could jeopardize the process through compromising integrity [22] and through information asymmetry [23]. Further, there is a danger that the RI trajectory might not only encounter obstacles but might also be directed against the RI aspiration and be perceived as a development barrier in traditional corporate cultures [24]. As such, the promise of RI could transfer to a challenge to innovative behavior. Therefore, some clues should be developed as to which stakeholders to include.

Implementation of responsible innovation on the firm or project levels requires a good fit between the purpose that aims at innovating for the benefits of society and the outcomes of innovation. In order to achieve this fit, inclusiveness seems to be a key element, which ultimately leads to better anticipation [11]. Further, adaptive capabilities are necessary to be able to reflect on feedback and respond to demanded changes in a responsible way. Thus, the process of participation and deliberate inclusion of relevant stakeholders becomes crucial during the whole innovation process.

Involving stakeholders makes decision-making processes more open and participatory, as well as more focused on sustainable development. The search for solutions demands the inclusion of stakeholders that opens up for new visions, purposes, questions, and dilemmas. This encompasses the collective deliberation through processes of dialogue, engagement, and debate, inviting and listening to broader perspectives of audiences and diverse stakeholders, and revolves around a quest for social legitimacy for innovation [25]. The main goal of inclusion is to diminish the authority of experts, with the inclusion of new voices in the governance of science and innovation as part of a quest for legitimacy [26]. This allows the introduction of a wide range of perspectives to reformulate issues and to identify potential contestation areas [8].

However, stakeholders' inclusion into the innovation process may also cause some challenges. Stakeholders might hold quite different and often conflicting views on the

problem in question, based on their diverse political, cultural, economic and social embed-dedness [27] Another critical point is that excessive inclusion may jeopardize the integrity of the common goods [22], as well as informational asymmetry [23]. Such conflicts can lead to slowing down the decision process [28], and consequently, the innovation process. Thus, for the economic agent, it becomes of crucial importance to recognize that inclusion at an early stage is a necessary condition to later acceptance and diffusion of innovation, but that this might also slow down the innovation process.

The question arises around the conditions under which inclusion is a necessary and useful tool. Moreover, who should be relevant stakeholders, and to what degree should they be involved in the innovation process—should they just provide inputs, or have a voice during the decision-making process? Lean models of innovation and agile method emphasize user inclusion at early stages and continuous interactions during the whole process [29,30]. The key characteristic of such methods is interactivity prolonged design space, which allow to "fail cheaply" and find alternative futures for solutions. Florèn et al. [31] argue that innovation project failures often occur due to too early lock-ins in the "fuzzy front end" of innovation, when dominant designs were established too early without enough inputs or experimentation. Thus, RI principles of anticipation, inclusion, and reflection fit well and add to well-accepted methods that allow flexibility and prolonged design space for innovation processes. Such methods have proved to be especially valuable under conditions of high uncertainty—both technological and in relation to market. Such high uncertainty is often associated with radical and disrupting innovations [32] and this suggests that the principles of RI can be useful in this context. In particular, in the context of digital health innovations, recent works point to the importance of keeping the design space open as long as possible, and on the possibilities of applying design thinking approaches [33,34].

## 3. Changing Role of Users in Digital Health

One area where this discussion has considerable relevance is in healthcare, not least because innovation involves a multi-layered decision-making process with clinical, man-agerial, and end-user (patient) levels involved in different degrees. Whilst there is extensive rhetoric around the importance of improving the patient experience and hearing patient voices, the extent to which this translates to actual implementation varies considerably.

The healthcare industry has long relied on traditional and linear models of innovation: basic and applied research followed by development and commercialization [35]. This is a technology-push model of innovation that has the aim of sustaining the current market players in the industry [36]. Healthcare is all about the patient—yet, for so much of its evolution, the patient has been treated as a passive recipient. Traditionally, patients have been a passive recipient of healthcare, and have been a victim of the circumstances rather than a powerful actor, leading to one-way communication relationship between healthcare professionals and patients [37]. Patients are often reluctant to assert their interests in the presence of clinicians, whom they see as experts [38]. As a result, many patients are prone to the alleged "hostage bargaining syndrome" (HBS), meaning patients behave as if negotiating for their health from a position of fear and confusion. It may manifest as understating a concern, asking for less than what is desired or needed, or even remaining silent against one's better judgment.

However, new digital technologies open up opportunities to change this situation. Digitalization of healthcare empowers patients to shape and direct the technologies in their own interests [39]. There is literature that acknowledges the fact that software becomes a medical device and that touches upon the role of patient in that process [40,41]. The new technologies give rise to the idea that healthcare should be all about the patient. New technology enables patients to gain a more active role in healthcare systems. As pointed out by Giustini [42], there is an increasingly vast volume of health information available to patients and doctors, and these health data can be effectively exploited and their presence can support the active involvement of the patient.

The rise of digital technologies in the healthcare sector leads not only to technological development, but also to a change in the state-of-mind, a way of thinking, an attitude towards healthcare, and role of different stakeholders in it [43]. This opens up opportunities for increasing participation of consumers in digital health outside of the hospital setting. Business intelligence and digitalization can become an important driver of patient-centredness [44]. Patients today are not only able to search for information with regard to their situation, they also become active discussants of their situation with healthcare professionals, and at some extremes they can become providers and a source of innovative solutions.

To the latter, there is a long tradition of "user-led" innovation in the sector and a significant proportion of this comes from patients (and their carers who develop and prototype solutions that benefit their situation). This is particularly relevant in those cases where the rarity of the disease or condition makes the market signaling weak for commercial healthcare providers.

It is possible to hypothesize a spectrum of patient involvement in innovation, ranging from the purely passive recipient of care right through to such user innovators at the other end. This disaggregation of the otherwise large but homogenous grouping of patients helps focus our attention on different types of patients and their potential inclusion in the innovation process. In particular, we might identify three positions along this spectrum:

(1) the "informed patient," equipped to use technology based on improved understanding;
(2) the "involved patient," playing an active role within a wider healthcare delivery system and enabled to do so by technology;
(3) the "innovating patient," providing ideas of their own based on their deep understanding of their healthcare issue.

Below we broadly discuss this three core groups of patients viewed through the lens of innovation.

### 3.1. The Informed Patient

Communication in medicine is a work in progress. We are immersed in data, directly delivered by the Internet into our homes. We can access reams of raw, un-spun research results; these turn up in government-funded, transparency-directed databases and in the online supplements to serious scientific journals [45]. At the other extreme, we can take in processed, neat-and-tidy bits of medical knowledge from commercials and TV documentaries. The Internet allows us to search and get answers to questions in a matter of seconds. This is of course true for the patients facing the questions on how to treat their illness. We can speak about the emerging concept of an informed patient. This means a patient who is knowledgeable about his or her condition and engaged in health decisions. This can ease the job for the doctor and bring more knowledge into the health system. The informed patient today has much better knowledge access in comparison to even a decade ago, with general search as well as specialized databases available to service information search. Better understanding of symptoms and treatments, and availability of feedback through for a and social media contribute to balancing the power in the health professional-patient relationship. As argued by recent research, there is new potential for information parity and equality and, as a result, emerging authority for patients to be the decision maker [46]. The informed patient might challenge the power of the relationship through questions, and obtain a better understanding about treatments, consequences, and choices available.

### 3.2. The Involved Patient

The development of a patient-centered approach to medicine is gradually allowing more patients to be involved in their own medical decisions. According to the predominant culture, research is performed on patients, not with patients [47]. Thus, patients continue to be regarded as a source of data and not as the true protagonists in the process. However, patient involvement is crucial for identifying the questions to ask and the outcomes to

assess [48]. Digitalization not only provides users with information; it also empowers them to share their feedback in a timely manner. A wide variety of feedback emphasized by digital technologies can be observed—from tele media communications with doctors and nurses to forum discussions on social media. This allows utilization of patients' feedback to the mutual benefit of healthcare provider and patient. This also makes it possible to improve and adapt products, services, or processes within a healthcare settings [49]. Recent studies demonstrate that patients' engagement contributes to increased innovation in medical units [50]. From being an informed patient, users now become co-creators in the innovation process through engagement. However, user feedback and ideas have to be heard and transformed into outcome. Thus, this co-creation process is dependent not only upon patients themselves, but also on the willingness of the organization (for example hospital) to absorb and react on this type of feedback. This requires responsiveness and reflectiveness to be in place to ensure the responsible innovation process [14,51].

Today it is increasingly common to involve patients or patient advocacy groups in study design. Patient advocacy groups now claim that their opinions must have greater influence on the decisions that affect them, which is reflected in the phrase "nothing about me without me." The development of new healthcare management models where patients become clients and the enormous expansion of information technology are additional factors that contribute to accelerating this change [50]. Patient-centered medicine cannot be practiced without patients participating in their own healthcare decisions and in the research that informs such decisions [52].

Although this cultural shift is beginning to change the way we understand healthcare, it is not having the same impact on the research process [53]. This may be because society does not see patient responsibility to participate in research as obvious as the responsibility to participate in their own medical care [54].

Vahdat et al. [55] found that an effective relationship of healthcare provider with patients is an important contributing factor of patient involvement in decision making. In addition, a literature review confirms that patients, in their journey through the healthcare system, have the right to be treated respectfully and honestly, and where possible, be involved in their own healthcare decisions. For patients' participation, mutual communication between the treatment team and the patient is necessary, so that information and knowledge could be shared between them, giving the patient a sense of control and responsibility, and thus involving the patient in care activities (mental or physical), to benefit and rehabilitate from this involvement [56,57].

### 3.3. The Innovative Patient

An alternative emerging at healthcare institutions worldwide is human-centered design and co-creation, a set of approaches that can accelerate and humanize healthcare innovation. This model is not just about getting greater patient feedback during the innovation process. Patients are co-designers, co-developers, and increasingly more responsible for their own and collective health outcomes. Canhão et al. [58] report on studies of patient- and caregiver-originated innovations. In their research they found a variety of ingenious solutions to daily problems, previously unknown therapies and treatments, and even new ideas for medical devices. Surprisingly, 8% of the total rare disease patients and caregivers had developed innovative solutions that even medical experts evaluated as novel. They concluded with that patients and caregivers around the globe may represent a tremendous source of knowledge on how to improve care.

In sharp contrast to the frequency of patient-to-patient sharing (88% of those who shared solutions), only 6% of patients reported describing their innovations to their clinicians, where reduced appointment time or lack of confidence are possible reasons [59,60]. Canhão et al. [58] claims that the absence of diffusion signify barriers to sharing as the inventors may lack time, skills, and opportunities to embark on the long process between idea and successful commercialization when the profit potential is limited or the inventors' motivations lie elsewhere. Simpler solutions may not be shared because the innovators

do not have contact with a wider community that would benefit. While some trade their innovations in patient support communities, others are less outgoing. They conclude that since all patients interact with the medical community, the latter can be essential in the distribution of new ideas. Physicians and other members of the care team could be attentive to innovations coming from their patients, and even active in reaching new patient groups which the innovation may benefit. Patients are not the only innovators, of course.

Summarizing, digitalization of healthcare services has led to a spectrum of patient behaviors, ranging from informed to involved and finally towards innovative users [61]. The informed patient is actively using sources of information to understand own circumstances. The involved patient is actively involved by the healthcare sector through sharing information and providing feedback on own conditions and healthcare routines or innovations. Finally, the innovative user is actively innovating and finding new, not previously exiting solutions for their own health problems. However, knowing that this spectrum of patient behavior exist is not enough. We need to look on the firm practices to understand better how those different behaviors can be captured and utilized into a firm's innovation process. In the next section, we describe practical cases in which we observed how firms in different countries, but within the same space of digital healthcare, are challenged with including user voices into their innovation process.

## 4. Method

Our study is based on the results from the major research program that includes eleven cases from six different countries—Norway, Portugal, USA, UK, Brazil, and the Netherlands (Research project "Digitalize or Die," granted by Norwegian Research Council for 2019-2020, project number 247716/O70). Data were collected in 2017–2018 and included interviews with firms operating in the digital health and their stakeholders. The cases are all deemed as typical cases in digital healthcare. Application of information and communication technologies on a wide scale—what we term "digital healthcare"—include robot surgery, telemedicine, electronic medical records, "smart" homes, "connected medicine," and much more (House of Commons, 2016), and how organization include users into their innovation process was the focus of interviews. Our cases represent a wide spectrum of technological solutions in that sector—from information-sharing platform "patient innovation" to digital health TV and specialized medical devises that apply sensor technology and digital communication. For the present article, we have focused solely on inclusion of users and their role in innovation process of the firms. It should be noticed that the data collection method for most of our cases were interviews; however, some cases have a more rich research history as they were followed by researchers over a period a time, and in such cases (for example cases A, D, E, see Table 1) materials are stemmed from a range of earlier publications, observations, surveys, as well as interviews. In such case, previous publications are used as references when we analyze cases. Moreover, detailed case description is available in the book, where each of the cases constitute a separate chapter [61].

Although cases are varying with regard to organizations in focus (cases C, D, G, J, and K are start-up companies; cases A and H are platforms; cases E and F are more established and bigger companies; case B is hospital; case I is university employee), they all are focusing on digital solutions in healthcare.

Lincoln, Y.S. et al. [62] recommend four criteria for evaluating interpretative research work: credibility, transferability, dependability, and confirmability. Consequently, the breath of the cases ensures triangulation of the data. To satisfy credibility criteria, all three authors discussed the cases among themselves. Eleven cases presented in the current study are from six different countries. This breadth and diversity of contexts allows to assume some degree of transferability in addition to dependability of the data. Interviews were either recorded and transcribed or researchers provided authors of this article with notes translated to English in case interviews were performed in other than English and Norwegian languages. Thus, this study satisfies the confirmability criteria.

**Table 1.** Table of cases.

| | Title of the Case | Description | Data Source |
|---|---|---|---|
| A | Patient Innovation | This case describes the web-based solution for sharing innovative solutions developed by patients or next of kin. It represents a non-profit organization, originally formed in Portugal. It became truly international and today around 1500 solutions from over 70 countries can be found on this platform | Based on eleven cases of patient innovations, as well as on the data from the web-based platform, surveys of web-page contributors over the period 2016–2020. |
| B | Patients feedback | This case is based on interviews with hospital employees in Norway. It sheds light on how patients provide feedback to healthcare professionals and how hospital, as organization, reacts to these feedback | Based on eight interviews performed 2019 with hospital employees (includes doctors, nurses and administrative staff) |
| C | My Chart | This case describes a web-based solution and app developed in University of Virginia, USA, to allow patients making appointments and obtaining important healthcare information from doctors | Based on two interviews performed in 2018–2019 with a doctor and IT specialist who were close to establishment of this app, as well as available additional materials (strategy documents, web pages, news in media) |
| D | Healthcare TV | This case describes a UK-based start-up, the main idea of which was to disseminate the specific knowledge, previously available only to doctors, to the general public through a series of short informational programs | Based on approximately ten interviews with five respondents, including the founders of the firm and stakeholders, performed during the period 2017–2019, as well as available additional materials (strategy documents, blogs and web pages, news in media) |
| E | Laerdal Medical AS | This case is around a Norwegian company founded in the 1940s and its transformation to achieve a major mission of saving lives. Company produces resuscitation equipment and today has managed to achieve its goal of saving over half a million lives per year | Based on three interviews with managers in the company, researcher, and daughter-company employee during 2017–2019, as well as additional materials such as web-pages, book issued by the firm itself, press-releases, and strategy documents |
| F | Blink case | Blink is one of the innovative projects of the Norwegian company Lyse AS. It implies an innovative video service concept for elderly people and at the moment of research was at the pre-launch stage of development | Based on personal observation and participation in the process. |
| G | Medicos (name changed) | A Norwegian start-up in digital health offers video consultations with doctors. They are a rapidly growing and developing company and offer a digital alternative to some public health services | Based on six interviews, three with three start-up companies and three with their stakeholders. Discussion on online fora and newspaper articles were used as supplementary materials |
| H | CareConnect (name changed) | This case focuses on a Dutch healthcare foundation, established as a conglomerate of over 400 organizations and institutions. It represents a platform where these actors can interact for the benefit of the patient. | Based on ten semi-structured interviews with key stakeholders in the development of the overall system in the summer of 2017 (seven company representatives and three stakeholders) |
| J | Hand Talk | Brazilian start-up company that uses avatar to translate and back-translate the deaf people's language. Company is launched and successfully developing | Based on cases of two Brazilian start-ups. For the first company, three founders and four stakeholders were interviewed; for the second company, founder, employee, and one stakeholder were interviewed in 2018–2019. |

**Table 1.** *Cont.*

| | Title of the Case | Description | Data Source |
|---|---|---|---|
| I | Academic entrepreneur | This case is focused on UK-based academic entrepreneur that worked on digitalization of healthcare records and immersive interacting learning tools. | Based on interview with academic entrepreneur performed in 2018. |
| K | WellStart Health | This is the case of USA-based start-up company that offers digital therapeutics—software-based interventions that help treat diseases by positively changing individuals' behaviors and closely tracking outcomes. It is on launch phase of development | Based on interview with founder of the company, company employees, and stakeholders, as well as on additional materials like web-page, press-releases, and strategy documents. Performed in 2018–2019. |

While analyzing cases, we focused on type of users (informed, involved, or innovative) and their role in innovation process of the above-mentioned firms.

## 5. Towards the Changing Role of Users

In the theory part of this article, we emphasized concepts of inclusiveness, anticipation, reflexivity, and responsiveness based on Stilgoe's [6] framework. In this part, we reflect on the inclusivity aspect and combine that with our view on changing role of patients that we have observed during our empirical work

### 5.1. Inclusiveness

Our cases confirm the view that digital health opens up significant opportunity for increasing participation of multiple stakeholders—including patients themselves. In almost all of our cases (ref Table 2), patients are recognized as an important potential knowledge source, and there is evidence that design methods are increasingly being used to ensure their input is captured at an early stage. We saw that degree of user involvement can be quite different and constitute a spectrum of involvement—from patients being informed, to involvement and co-creation, and toward the innovative patients. The table below provides evidence of the type of patients and their role in the innovation process based on the empirical cases, followed by the detailed discussion of each type of patient and their involvement within the cases. It should be noticed that several cases can be placed in both "informed" patient and "involved" patient categories. In the Table 2 below we, however, placed our cases just in one category. The cases were put into "involved" category when the degree of interaction and user involvement is important for application to be functional.

**Table 2.** Analysis of cases: type of patient and their role in the innovation process.

| Type of Patient | Description | Cases | Role in Innovation Process |
|---|---|---|---|
| The Informed Patient | Patient is actively using sources of information to understand own circumstances. | C (My Chart), D (Healthcare TV), B (Patient Feedback), K (Well-Start Health), G (Medicos) | Feedback on information access |
| The Involved Patient | Patient is actively involved by the healthcare sector and their feedback is necessary for solution to be functional. | B (Patient Feedback), H (Care Connect), F(Blink), J (Hand talk), I (Academic entrepreneur), E (Laerdal) | Reciprocal information exchange, feedback on experience |
| The Innovative Patient | Patient is actively innovating to find new solutions for their problems. | A (Patient Innovation) | Patients are the main drivers of the innovation process |

### 5.2. The Informed Patient

At the minimum the scope for the informed patient—a patient who is knowledgeable about his or her condition and engaged in health decisions—is emerging. This can ease the job for the doctor and bring more knowledge into the health system. This is illustrated well in the case called "MyChart" example from USA, where patients have better information access to their health records and communication with healthcare institutions through an app or web-page. Similar Electronic Health Records systems have also been implemented in the Norwegian hospitals studied.

Digital innovation can also help inform and empower patients through improving access to information and this was the original idea behind a UK case, called "Healthcare TV." Although the innovation itself came about through the actions of a "hero" innovator, developing a prototype on the back of the poor experience of his father, the desire to create a more open easy access way of informing patients has considerable potential. Significantly, the case shows the conflict between the objectives and interests of the clinical audience and the patient audience; in the evolution of the business model, the innovation pivoted to create two complementary but different video platforms.

As demonstrated in case "Patient feedback" from Norway, patients do provide a lot of feedback to clinicians. However, reflecting on the feedback requires system changes, which are not always supported. Patient feedback enhanced by digitalization for the sector might bring considerable value, enhancing measurable outcomes such as quality of care and availability of care.

Another US-based case, "WellStart Health," demonstrates that this application can only function by reciprocal information exchange with patients. These cases suggest a new dynamic in healthcare—an open, modern discussion between patients and physicians. Professionalism is the basis of medicine's contract with society, and the principle of patient autonomy is a fundamental value of the medical profession. This allows for a stronger patient autonomy, empowering patients to make informed decisions about their health.

### 5.3. The Involved Patient

Most of our cases revolve around the "involved patients," who are able to be co-creators and contribute to the innovation process at both the design and diffusion ends of the journey. Several cases illustrate the situation of the involved patient. Case "Patient feedback" demonstrates that such patient feedback in many cases can stimulate organizational improvements and service innovations in hospitals. Using the particular example of a detailed longitudinal case of the development and diffusion (with subsequent modification and "re-innovation") of a digital health information platform in case "Healthcare TV" identifies a number of key points at which the innovation concept "pivoted" to reflect new information, some of which resulted from a wider level of inclusion. Co-creation of digital innovative solution for elderly citizens by different stakeholder groups, including relatives, volunteers, and healthcare professionals, is described in case CareConnect. This case demonstrates a bottom-up initiative from users to create a platform to allow better services for elderly people in the Netherlands.

A priori the case for involving patients is strong, not least because they help in identifying the questions to ask and the outcomes to assess [44]. One of the valuable features of digital technologies is the possibility of embedding such insights at an early stage; since the technologies are programmable and re-programmable, there exists scope for co-creation during the development phase. Such approaches have increasingly been used in improving processes and services, drawing on design thinking tools to enable a better understanding of "the patient journey" and adapting delivery systems to those insights [63,64].

However, user feedback and ideas are only useful if they are transformed into outcomes. Thus, this co-creation process is dependent on three factors: (1) the ability and willingness of patients to get involved and contribute their insights, (2) the willingness of the innovating organization to absorb and react to this type of feedback, and (3) the flexibil-

ity of the technology—the degree to which it can be shaped and modified in response to patient insights.

Clearly, service and process design are more amenable to such in-line modification; product design is often less flexible. However, as we noted above, a key feature of digital technologies is their inherent flexibility and the potential to "delay the freeze" in terms of final design and production at scale.

To some extent, there is a dynamic visible in our case studies that drives towards greater patient involvement because it leads to better design and downstream outcomes. The "Academic entrepreneur" case from UK is a good illustration; here, the original approach was to a large extent driven by interactions with clinicians. However, experience with users, especially feedback from an asthma patients, led to revisions not only of that specific design but of the subsequent approach taken by the innovator.

One of the important aspects of working with involved patients is that they provide a helpful coalition through which to learn and develop novel technologies. The example of digital therapeutics "WellStart Health" from USA is a good example; here, the opportunity to work with something that has considerable potential as a disruptive innovation is enhanced by an active involved user base. Christensen's prescription in such situations is not to work with the mainstream but rather to find ways of exploring at the fringes—and gaining valuable experience by working with users [32].

So, the question is less one of whether patients have the knowledge than about how to enable its articulation and transmission at an early enough stage in the design process. A variety of studies [65,66] suggest that design thinking and methods provide a rich toolbox and there is evidence amongst our cases of using this kind of approach. There is probably scope for more—and in the process we might well be able to move patients from the passive to the active end of the co-creation spectrum by giving them voice.

In the "Blink" case, we can see the power of such approaches. Here, technology developed for elderly people required a few interactions with users at early design stage and throughout the whole design and test stages to ensure user-friendliness of the application. Even though there is careful planning and attention to user needs at the front end, there is still scope for unexpected issues to emerge. The methodologies in use by the company allowed for an "early reveal" of problem issues that could then be dealt with in a revised version of the innovation—essentially the above principle of "pivoting" in practice.

One major challenge, especially for start-ups lies, in their intersection with the wider healthcare establishment. Most of our start-ups recognize the potential value in working with users. Brazilian start up Hand Talk, USA-based start-up Well Start Health, and Norwegian-based start-up Medicos generally had a higher level and an explicit concern for patient inclusion. However, sometimes their experience, as illustrated in case Medicos, is that they often come up against health systems that are closed to them and where their ability to bring in RI solutions is to some extent constrained by institutional walls.

RI capability can evolve over time. For example, the Laerdal company began with a strong sense of purpose linked to the RI agenda. Founded in response to a crisis event (the near drowning of the founder's son), the subsequent development of enabling products and services to help save lives gives an important social mission to the business. The idea of widespread inclusion of different perspectives was another key foundation; the need for bringing in a wide variety of expertise drew in many medical experts to help develop the first product, the Resiscu Annie mannequin back in the 1960s. Since then, many changes have occurred, but the core principles of the company remained, drawing on two key tributaries, networks of medical knowledge and close user engagement. Today, Laerdal is a global company with over 1500 employees in 24 countries; its core mission remains one dedicated to saving lives through resuscitation, emergency care, and patient safety. Embedding RI principles of the kind we have been looking at have helped the business grow while remaining true to these core social values. As the founder Aasmund Laerdal put it:

*" . . . . . . sustainable business is only possible through developing ability to listen, endless curiosity, practical problem solving, respect for customer, hard work and a passion for continuous improvement"* [67] (p. 3).

The challenge now comes in trying to extend and replicate these values across an increasingly diverse set of partners as the company moves to a more "open innovation" set of collaborations with external agencies.

*5.4. The Innovating Patient*

This type of patient has been made visible through the "Patient Innovation" case, which describes a web-based platform for sharing innovative solutions from patients and next of kin. Over 650 innovations developed by patients and caregivers are posted on the Patient Innovation website. Some of them are technically very simple, but offer great value to patients and their families, while others are more advanced and complex. Most of them focus on increasing patients' autonomy and quality of life. Evidence also shows that these innovations have low diffusion rate. This is, especially, a situation in the segment of rare diseases and chronic needs—niche markets, unattractive to stakeholders in the healthcare industry—where patients are pioneers regarding innovation when compared to commercial producers, which can help close gaps in the delivery of medical care [68].

As von Hippel and colleagues have demonstrated over many years, users are a powerful source of insight at the front end of innovation [69]. In particular, a combination of their high incentive to innovate and their tolerance for imperfection means that early stage prototypes often open up radically new directions for innovation—and because of their origin also open up a wide potential downstream marketplace for "people like us." Diffusion is linked to compatibility and being able to meet precisely the needs of users is a powerful accelerator for adoption.

Much of this depends on the articulation of "sticky information"—the tacit knowledge that may not be evident to external agents nor even to advanced market research techniques [70]. Users know, by definition, what it is they want and the conditions under which it will work—the challenge is finding ways of capturing and working with that.

User-driven innovation is a potentially rich resource—and nowhere more than in healthcare. Who knows the challenges of leaving with diabetes or cancer better than those who experience it? Patients with chronic diseases are often forced to find solutions for their conditions, if solutions for their problems are not available already on the market.

Many people, across all fields, innovate to address their unmet needs, in a model that MIT researcher Eric von Hippel calls "user innovation." It builds on decades of user-led innovation research and is predicated on a new or different model of the innovation process [69,71,72]. The traditional model sees an innovating organization—perhaps an R&D lab, or a small team of clinician experts—developing solutions for a target market of sufficient size to justify the risky exploratory research. Without signals of the likely market potential, this research will not happen, effectively closing down many potentially interesting avenues before they have even been started. By contrast, the user-led model, as we see, is about self-interest and immediate peers [68]. The innovation incentive is solving a problem; if there are useful spillovers that others can benefit from, then this is a bonus, but the original innovators are content with solving their own problem. There is no incentive or desire to move to scale. The potential of the "free innovation" model is to bring these two worlds together by: (1) mobilizing the front-end risky exploration by capitalizing on user insights, sticky knowledge, etc. and early prototyping across tight but small peer communities; (2) linking that to the "productionizing" and marketing expertise of the mainstream innovation system, which is capable of scaling up once initial trajectories are established.

This is a potential win-win collaboration and one which is seeing increasing interest in the commercial sector. Evidence of this is found in the Patient Innovation case using the example of a shirt developed by breast cancer patient that was patented, approved by the FDA as a Class 1 medical device, and commercially available in 36 countries. Here, close

user engagement especially with pioneer and hero innovators at the extreme end of our spectrum could provide early-stage input to innovations which could be developed and scaled for much wider audiences. What is needed is the mechanisms to bring these two worlds closer together.

There are opportunities here for the healthcare system to become active partners in this process—in which they are able not only to get improved quality design but also downstream acceptance and diffusion. This is potentially rich territory where taking an inclusive RI approach can pay dividends.

## 6. Conclusions

To summarize, responsible innovation matters—of course. However, there is a need to convert it from a "Motherhood/apple pie" slogan to a practical approach, one whose benefits are clear and which encourage innovators to adopt it. The idea of the inclusion of multiple stakeholder views and insights is of interest; whilst it is difficult and time-consuming to do so, the growing evidence is that engaging with this perspective can enhance design and diffusion. In parallel with the experience of the quality movement back in the 1980s, when the perception of quality management as a necessary but undesirable cost was transformed to a realization that it was in everyone's interest to take on the concern for quality and to build it in at the source [73], in similar fashion RI might become a widespread way of understanding the innovation process.

### 6.1. Theoretical Implications

To enable RI, there is a need for a framework and guidelines around it to build such innovation. The Stilgoe et al. [14] model helpfully identifies four key dimensions, which offered a template for RI design. However, it can be argued that, taken alone, the framework might be seen as a little prescriptive and static. What is missing is some way of looking at RI as being enacted through a process over time. Making RI happen is about managing the contestable nature of innovation—its trajectory is always a product of social shaping forces. For any innovation, there is design space. It is wide in the early fluid phases of a technological field but there is also still scope for moving the walls of an established trajectory. However, as the dominant design emerges and a technological trajectory becomes established, so this room to maneuver becomes limited and the choice constrained. Extending the discussion of Stahl's maturity model [16], dimensions of RI, and the refined [74,75] framework, our exploratory cases suggest that there is value in maintaining a wider more inclusive approach and "delaying the freeze" in terms of innovation design can offer significant advantages in terms of quality of design and user acceptability.

RI literature, although acknowledges the diversity of stakeholders [15,20], does not touch upon the spectrum of stakeholders innovative engagement and their role in innovation process of the firm. Our empirical evidence suggests that user contribution into innovative process in organizations would vary depending on weather they are informed, involved, or innovative users.

### 6.2. Policy Implications

As we identified in this study, one area where there is considerable scope for opening the design space is in enhancing the role of users. Users are demonstrably a valuable source of insight and diffusion accelerators, and evidence suggests we should make more of them. There is a scope for policy intervention to create a more favorable context for inclusion by enhancing the role of users, empowering and engaging patients more extensively. Policy options here might include training patients to bring them further along the involvement spectrum, introducing mechanisms that give voice to their insights within innovation processes and using procurement policy to help set and shape the direction of such activity, and privileging innovations that can demonstrate a high degree of user engagement. Regional and national initiatives can include the development of organizational networks,

clusters, and conglomerates that aim to make user inclusiveness less time- and resource-demanding for each individual economic actor. There may also be scope for innovation lab type environments enabling co-creation and facilitation of entry to the mainstream system.

However, in practice there are multiple obstacles—some of which are more susceptible than others to policy intervention whether at state level, where procurement and reimbursement regimes give the state significant shaping power, or at the enterprise level. For instance, several of our cases illustrated a problem which we might term the "institutional wall" effect—that is, significant resistance to innovations perceived to be radical and challenging existing structures or routines. For example, the established healthcare system is often uncomfortable in dealing with patient initiated innovations and this "not invented here" problem may require considerable additional effort to overcome such inertia. The Medicos case illustrates that because the healthcare context often is heavily politicized with government involvement and large, powerful companies, it is often difficult for innovating patients and other start-ups within this sector to break through this institutional wall.

There is scope for policy intervention to help deal with this, whether in terms of institutional strategies or at a wider policy level. For example, the Academic entrepreneur case showed that the university system may be an important actor in developing innovations and policy interventions can further stimulate such actors.

Regulative, cognitive, and normative constrains may prevent existing systems from fully integrating this potential. Therefore, we could imagine a new healthcare system where the institutional wall is made more flexible through an innovation system that is geared towards these kind of start-ups.

## 7. Future Research

It is worth concluding with some points that have been raised but not explored in this article. These might well constitute a valuable future research agenda in this important field. We would highlight six key themes: (1) We have stressed the importance of user engagement and the potential contribution that this offers, but more needs to be done to understand who the agents are that might orchestrate stakeholder's participation. (2) Typically, development of products/services is considered a complex process that requires the management of several factors at different stages, running from concept, through project design and testing to product launch and marketing. Thus, the question relates to improving our understanding of when (at which stage of the innovation process) users might most helpfully contribute. (3) Innovation can take many forms. For example, product innovation (change in products/services offered by a company), process innovation (change in the way products/services are offered or presented to the consumer), innovation of position (change in the context in which the products/services are introduced in the market), and paradigm innovation (change in the basic mental models that guide the actions of the company). This raises a third question: how can stakeholder's participation contribute to innovation in these different forms? (4) A fourth question relates to the "technology" of user participation. There are many useful frameworks and methods emerging like the Business Model Canvas, but more needs be done to look systematically at which tools and frameworks are relevant for which stages and for different groups of users. (5) We have focused on users and patients in particular, but the RI argument is about widening stakeholder involvement and so there is work to be done around mapping stakeholders and understanding the issues which may emerge in working across a diverse group. These might include conflict of interests, fear of loss of power over the process, fear about the relationship between secrecy and transparency, as well as operational aspects such as time consumption and other resources. We also need to recognize that innovations do not happen in a vacuum. They are embedded in particular contexts—country or region—and we need to take the influence of these into account. Responsible innovation raises questions of broader inclusion, not only of local stakeholders, but also a wider set of views and perspectives, for example around gender or race inequality and about inequality of knowledge. This raises an even broader question of how to make innovative solutions

responsible across countries, for societies with quite different knowledge bases and value systems? Would inclusion of only local stakeholders into the innovation process ensure globally responsible outcomes? (6) Finally, we mentioned that user innovation diffusion has a limited scope. This is particularly the case user-initiated innovations in the healthcare sector. Future research might address the reasons for that and try to find solutions for a better diffusion for such innovative solutions.

Disruptive innovations, like those in the field of digital healthcare have the potential to change and challenge established systems, and so it is important to ensure they are designed and diffused in a responsible way. In this article, we have tried to show that principles of RI are not something that is required out of philanthropy, but rather that exercising responsible innovation might enhance successful commercialization. This provides a hope that recognition of the inclusion, anticipation, and reflection principles will become a natural part of accepted innovation practice by economic actors.

**Author Contributions:** Conceptualization, T.I., E.O., and J.B.; methodology, T.I., E.O., and J.B.; formal analysis, T.I., E.O., and J.B.; writing—original draft preparation, T.I.; writing—review and editing, T.I. All authors have read and agreed to the published version of the manuscript.

**Funding:** This research was funded by Norwegian research Council, grant number 247716/O70 and grant number 299192, and the APC was funded by University of Stavanger.

**Institutional Review Board Statement:** Not applicable.

**Informed Consent Statement:** Not applicable.

**Acknowledgments:** Authors are thankful for all researchers that provide cases materials for this article and for earlier publications from the projects that funded this article.

**Conflicts of Interest:** The authors declare no conflict of interest. The funders had no role in the design of the study; in the collection, analyses, or interpretation of data; in the writing of the manuscript, or in the decision to publish the results.

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
