# Peer review of "Changing Role of Users—Innovating Responsibly in Digital Health"

_sustainability, doi:10.3390/su13041616_

Round 1
Reviewer 1 Report
This article is incomplete and lacks a full list of references. For this reason, I cannot review and it should be sent back to authors to complete and resubmit.
Author Response
Dear Reviewer
We have now improved the article on several fronts and made a full reference list. We hope you can consider this article for further review
Best regards
Authors
Reviewer 2 Report
Dear Author(s), attached some elements to strengthen the paper.
Title and Abstract
Relevant and informative. Probably, I would suggest to include implications in the abstract.
Introduction
Should be extend considering better why is important this analysis. The GAP is not so clear now. Additionally, the section needs to be extend considering the main implications obtained from your study and a brief map of the paper.
Literature review
The literature review is adequate in my opinion.
Method
I would suggest strengthening the method. Readers should understand why you selected those cases, why are so important? How do you select those elements? What do you use for the analysis? How do you perform the data collection? Through interviews? Data analysis?
The paper needs to strengthen the method and have a solid scientific framework.
Results
Due to the lack of the method, it is extremely difficult to understand the results. They are interesting, but it is difficult to understand how and using which process do you obtain those elements?
Conclusions
The conclusion has all the section needed. The main concern is for the method and results section which not have scientific soundness.
All the best!
The Reviewer
Author Response
Title and Abstract
Relevant and informative. Probably, I would suggest to include implications in the abstract.
Thank you for that suggestion. We have now included implications into the abstract.
Introduction
(R) Should be extend considering better why is important this analysis. The GAP is not so clear now. Additionally, the section needs to be extend considering the main implications obtained from your study and a brief map of the paper.
(Reply) Thank you for your suggestions. We have outlined implications in abstract and shortly in introduction. In introduction we have now clearly defined a gap in the literature the paper addresses and presented the further outline of the paper.
Literature review
The literature review is adequate in my opinion.
Method
(R) I would suggest strengthening the method. Readers should understand why you selected those cases, why are so important? How do you select those elements? What do you use for the analysis? How do you perform the data collection? Through interviews? Data analysis?
The paper needs to strengthen the method and have a solid scientific framework.
(Reply) Thank you for that suggestion. This paper is a part of the major international research project and our intention was to show diversity of cases of responsible innovations in digital health across different countries. We have now stressed this point in the method section. We also underlined what cases have in common. We have now specified the data collection methods for each case, by adding a new column “data source” to the table in method section. We explained that data analysis was performed for each case through lenses of inclusion, and with reference to three types of patients (informed, involved or innovative) identified in the theory part of the paper.
Results
(R) Due to the lack of the method, it is extremely difficult to understand the results. They are interesting, but it is difficult to understand how and using which process do you obtain those elements?
(Reply) We have improved the method section now. Hopefully this helps the reader to understand how we arrived into our results. The major logic was to analyze cases focusing on inclusion as the main term and also relate inclusion to three major types of patients (informed, involved and innovating) identified.
Conclusions
The conclusion has all the section needed. The main concern is for the method and results section which not have scientific soundness.
All the best!
Reviewer 3 Report
Dear Authors and Editor,
I am grateful for the opportunity to review this interesting paper concerning the role of users in the innovation process within the digital health context.
My main concerns about the manuscript are the followings:
- The article needs to be revised from a stylistic point of view. There are many spelling errors, type errors and misspelled words, as well as phrases to be completely rewritten. This is due to an inadequate level of English, so I strongly suggest the help of a mother tongue speaker to review the paper. Moreover, the bibliography of the article does not meet the standard of the journal: in line 27 and line 30 there are citations in which the author's name appears instead of the numerical reference and the bibliography needs to be completely revised because most of the citations are incomplete and do not make it possible to trace the original article. Finally, check the structure of the paper and correct the paragraph numbering.
- Line 149, the phrase "There is an increasingly vast volume of health information available to patients and doctors, making it easier than earlier for patients to develop their own diagnosis" may be misleading. Formulated in this way it suggests that the presence of health data allows the patient to develop its own diagnosis. It would be appropriate to discuss further how an important amount of health data can be effectively exploited and how their presence can support the active involvement of the patient. In addition, since the authors have cited new technologies able to put the patient at the center (see line 147, 148), it would be appropriate to mention some examples. I suggest to see the following work: Zheng, W., Wu, Y. C. J., & Chen, L. (2018). Business intelligence for patient-centeredness: A systematic review. Telematics and Informatics, 35(4), 665-676.
- The authors recognize three types of patients: informed patient, involved patiend and innovating patient, and explain the characteristics of each of them through the lens of innovation. However, from the literature proposed by the authors for the first type, the informed patient, it is not clear how he can contribute to the innovation process. Moreover, in Table 1 it would be appropriate to add an innovation column in which the role that each type of patient has in the innovation process is highlighted.
- The methodology section needs to be improved. First of all in paragraph 3 some hypothetical research questions are presented, called "questions to be addressed". It is necessary to give more space to research questions by legitimizing them with literature. In addition, an explanation of how the case studies were constructed can be added.
- The discussion of the case studies should be based on the proposed research questions and structured to give them an answer based on the evidence from the case studies. As for Table 2, since the case studies are not homogeneous (some are startups, some are well established companies, some are individuals or hospitals), I suggest rethinking its design by adding the column for the type of organization. In addition, since the entire article is focused on the role of the user, the authors can add a column where this is specified. In fact in some cases the role of the user can be deduced from the short description (see case B), while in others not (for example the case E). Finally, for each case the authors can specify whether it tackles the issue of informed, involved or innovating patient.
- Although the issue of diffusion/communication of the innovative solutions proposed by patients has been addressed by the case studies, it deserves further theoretical investigation to assess the barriers that prevent this from happening. This could be the subject of future researches.
Good luck with your work.
Author Response
Dear Authors and Editor,
I am grateful for the opportunity to review this interesting paper concerning the role of users in the innovation process within the digital health context.
My main concerns about the manuscript are the followings:
- The article needs to be revised from a stylistic point of view. There are many spelling errors, type errors and misspelled words, as well as phrases to be completely rewritten. This is due to an inadequate level of English, so I strongly suggest the help of a mother tongue speaker to review the paper. Moreover, the bibliography of the article does not meet the standard of the journal: in line 27 and line 30 there are citations in which the author's name appears instead of the numerical reference and the bibliography needs to be completely revised because most of the citations are incomplete and do not make it possible to trace the original article. Finally, check the structure of the paper and correct the paragraph numbering.
(Reply) Thank you for that concerns. We have now improved English in our paper, and we ensured all references are correct and are in accordance with the Journal standard. On lines 27 and 30 we did only found references to Lund Declaration and European Commission – we have inserted now number instead.
- Line 149, the phrase "There is an increasingly vast volume of health information available to patients and doctors, making it easier than earlier for patients to develop their own diagnosis" may be misleading. Formulated in this way it suggests that the presence of health data allows the patient to develop its own diagnosis. It would be appropriate to discuss further how an important amount of health data can be effectively exploited and how their presence can support the active involvement of the patient. In addition, since the authors have cited new technologies able to put the patient at the center (see line 147, 148), it would be appropriate to mention some examples. I suggest to see the following work: Zheng, W., Wu, Y. C. J., & Chen, L. (2018). Business intelligence for patient-centeredness: A systematic review. Telematics and Informatics, 35(4), 665-676.
(Reply) Thank you for this valuable comment. We agree with it and we have now clarified what we mean by phrase on line 149 (please note that line number has changed). Further, we studied the suggested article and integrated some valuable insights from it into our paper
- The authors recognize three types of patients: informed patient, involved patient and innovating patient, and explain the characteristics of each of them through the lens of innovation. However, from the literature proposed by the authors for the first type, the informed patient, it is not clear how he can contribute to the innovation process. Moreover, in Table 1 it would be appropriate to add an innovation column in which the role that each type of patient has in the innovation process is highlighted.
(Reply) Thank you for that concern and suggestion. We have now clarified the contribution of informed user in innovation process. We removed table 1 but inserted some text instead. Additionally, we have added a new table in the analysis section emphasizing how each patient type contributes to innovation process based on our empirical findings
- The methodology section needs to be improved. First of all in paragraph 3 some hypothetical research questions are presented, called "questions to be addressed". It is necessary to give more space to research questions by legitimizing them with literature. In addition, an explanation of how the case studies were constructed can be added.
(Reply) Thank you for this suggestion. Our major research question is presented in introduction, line 65, “how to empower users to become a part of innovation process in a responsible way?” We also outline that we aim to look on firm practices for user inclusion and reflect on three type of users identified earlier in the literature. “Questions to be addressed” were a part of table 1, which was taken from previous studies, and we have removed this table now and inserted some text instead. In the previous studies, that were questions for future research.
- The discussion of the case studies should be based on the proposed research questions and structured to give them an answer based on the evidence from the case studies. As for Table 2, since the case studies are not homogeneous (some are startups, some are well established companies, some are individuals or hospitals), I suggest rethinking its design by adding the column for the type of organization. In addition, since the entire article is focused on the role of the user, the authors can add a column where this is specified. In fact in some cases the role of the user can be deduced from the short description (see case B), while in others not (for example the case E). Finally, for each case the authors can specify whether it tackles the issue of informed, involved or innovating patient.
(Reply) Thank you for this valuable suggestion. The type of firms is specified the text in the method section. We have also added new table into analysis section, where we focused on type of patients identified for each firm as well as on their role in innovation process of each case. We hope that this table helps to get a better overview of the cases and provides add to clarifying our answer to the major research question of how to empower users to become a part of innovation process in a responsible way.
- Although the issue of diffusion/communication of the innovative solutions proposed by patients has been addressed by the case studies, it deserves further theoretical investigation to assess the barriers that prevent this from happening. This could be the subject of future researches.
(Reply) Thank you, we agree that this is important avenue for future research and have included this as a suggestion in “future research” section of the article
Good luck with your work.
Reviewer 4 Report
Thank you for inviting me to review this article.
The theme of the article is of interest, however the article is very supeficial.
It does not explore the case studies under analysis, not in terms of analysis neither in terms of results.
The methods, data analysis and results need to be reformulated.
Author Response
We have now considerably reviewed the article. We have focused on improvement of methods and describing analysis in more details., linking theory and research question to our empirical findings. We have emphasized that this paper is a part of the major international research project and our intention was to show diversity of cases of responsible innovations in digital health across different countries. We have now stressed this point in the method section. We also underlined what cases have in common. We specified the data collection methods for each case, and how we analyzed data. The major logic was to analyze cases focusing on inclusion as the main term, relating three major types of patients (informed, involved and innovating) identified to their role in innovation process of the cases.
We hope you can consider this article for further reviewer
Round 2
Reviewer 1 Report
Many thanks for the opportunity to review this work. I think the work has improved significantly if compared to the earlier submitted version.
The overall article reads well and rather than provide feedback on grammar/typos which should be picked-up on with a proofreading prior to the final submission, I want to highlight a red flag for me as I read through the work with great interest. Specifically, the article describes responsible innovation and developing a more inclusive approach to health care solutions. Admittedly, I was expecting to see a body of work on Design Thinking and how technology solutions are aligned to user needs - but this never came. For example, one publication that comes to mind is (but you'll find a few more with a quick search):
- Carroll, N. and Richardson, I., 2016. Aligning healthcare innovation and software requirements through design thinking. In 2016 IEEE/ACM International Workshop on Software Engineering in Healthcare Systems (SEHS) (pp. 1-7). IEEE.
- Kim, S.H., Myers, C.G. and Allen, L., 2017. Health care providers can use design thinking to improve patient experiences. Harvard Business Review, 95(5), pp.222-229.
I know that the above work focuses on e-health to support identify healthcare requirements. In addition some of the same authors went on to publish an article on the role of regulation in shaping responsible innovation (within a healthcare context) - among others (but a quick search will highlight 4-5 key references):
- Crumpler, E.S. and Rudolph, H., 1997. FDA software policy and regulation of medical device software. Food & Drug LJ, 52, p.511.
- Carroll, N. and Richardson, I., 2016. Software-as-a-Medical Device: demystifying Connected Health regulations. Information Technology, 18(2), pp.186-215.
My main comment is that there is a large body of knowledge around design thinking and digital health regulation and some (not all) of this should be referred to in your main body of work. I understand that the authors cannot address all of these various angles but simply acknowledge them and highlight such topics as future research also. Otherwise, you seem to describe related works but don't touch on the key areas, such as "design thinking", and "software-as-a-medical device" - especially when presenting work around responsible innovation.
I also enjoyed the methods section and contribution section One suggestion would be to add 'practitioner contributions' and how solution developers would use your work. The theory could also be expanded but I think the policy implications are sufficient.
Well done. I think this work makes for an interesting contribution to a number of areas including e-health, innovation, and software engineering. Perhaps, one of the issues is that the contributions section undersell the research implications. I also think if you add a little around the design think and software regulations as suggested, it will make for a very good publication.
Author Response
Reviwer 1
Many thanks for the opportunity to review this work. I think the work has improved significantly if compared to the earlier submitted version.
The overall article reads well and rather than provide feedback on grammar/typos which should be picked-up on with a proofreading prior to the final submission, I want to highlight a red flag for me as I read through the work with great interest. Specifically, the article describes responsible innovation and developing a more inclusive approach to health care solutions. Admittedly, I was expecting to see a body of work on Design Thinking and how technology solutions are aligned to user needs - but this never came. For example, one publication that comes to mind is (but you'll find a few more with a quick search):
- Carroll, N. and Richardson, I., 2016. Aligning healthcare innovation and software requirements through design thinking. In 2016 IEEE/ACM International Workshop on Software Engineering in Healthcare Systems (SEHS)(pp. 1-7).
- Kim, S.H., Myers, C.G. and Allen, L., 2017. Health care providers can use design thinking to improve patient experiences. Harvard Business Review, 95(5), pp.222-229.
Reply: Dear Reviewer
Thank you for that comment. We agree that design thinking is one of the important approaches that can help considerably to involve users into the innovation process, also in healthcare context. In our article we had at least two references to this method:
- Kamper, S. J., Maher, C. G., & Mackay, G. Global rating of change scales: a review of strengths and weaknesses and considerations for design. Journal of Manual & Manipulative Therapy, 2009, 17(3), pp.163-170. [CrossRef]
- Bessant, J., & Maher, L. Developing radical service innovations in healthcare—the role of design methods. International Journal of Innovation Management, 2009, 13(04), pp.555-568. [CrossRef]
We are grateful you suggested additional literature. We have reviewed suggested articles and added some insights from both suggested articles into the text now (and refences are added into reference list as well)
I know that the above work focuses on e-health to support identify healthcare requirements. In addition some of the same authors went on to publish an article on the role of regulation in shaping responsible innovation (within a healthcare context) - among others (but a quick search will highlight 4-5 key references):
- Crumpler, E.S. and Rudolph, H., 1997. FDA software policy and regulation of medical device software. Food & Drug LJ, 52, p.511.
- Carroll, N. and Richardson, I., 2016. Software-as-a-Medical Device: demystifying Connected Health regulations. Information Technology, 18(2), pp.186-215.
My main comment is that there is a large body of knowledge around design thinking and digital health regulation and some (not all) of this should be referred to in your main body of work. I understand that the authors cannot address all of these various angles but simply acknowledge them and highlight such topics as future research also. Otherwise, you seem to describe related works but don't touch on the key areas, such as "design thinking", and "software-as-a-medical device" - especially when presenting work around responsible innovation.
Reply: Thank you very much for pointing on relevant literature. We gave reviewed and added some insights from the suggested articles, but also, as suggested, we found more literature and referred to it in our article, especially in the conclusion section, such as:
Silva, H. P., Lehoux, P., Miller, F. A., & Denis, J. L. (2018). Introducing responsible innovation in health: a policy-oriented framework. Health research policy and systems, 16(1), 90.
Ribeiro, B., Bengtsson, L., Benneworth, P., Bührer, S., Castro-Martínez, E., Hansen, M., ... & Shapira, P. (2018). Introducing the dilemma of societal alignment for inclusive and responsible research and innovation. Journal of responsible innovation, 5(3), 316-331.
I also enjoyed the methods section and contribution section One suggestion would be to add 'practitioner contributions' and how solution developers would use your work. The theory could also be expanded but I think the policy implications are sufficient.
Reply: Thank you for this suggestion. We have added now “Practitioner contribution” to the conclusion section of our article.
Well done. I think this work makes for an interesting contribution to a number of areas including e-health, innovation, and software engineering. Perhaps, one of the issues is that the contributions section undersell the research implications. I also think if you add a little around the design think and software regulations as suggested, it will make for a very good publication.
Reply: Thank you
Reviewer 2 Report
Dear Authors,
Thank you for your modifications.
The paper now has several improvements. Thank you for your work.
However, I would suggest the authors read all the paper carefully. There are missing words. Therefore, moderate English changes are required.
All the best.
The Reviewer.
Author Response
Dear reviewer
Thank you for your feedback. We have re-read the article and improved when necessary.
Many thanks for your comments, they were very helpful in improvement of this work.
Reviewer 4 Report
The authors addressed to all my concerns.
Author Response
Reviewer 4
The authors addressed to all my concerns.
Thank you